# The Expression of Active CD11b Monocytes in Blood and Disease Progression in Amyotrophic Lateral Sclerosis

**DOI:** 10.3390/ijms23063370

**Published:** 2022-03-21

**Authors:** Ozlem Yildiz, Johannes Schroth, Vittoria Lombardi, Valentina Pucino, Yoana Bobeva, Ping Kei Yip, Klaus Schmierer, Claudio Mauro, Timothy Tree, Sian Mari Henson, Andrea Malaspina

**Affiliations:** 1Centre for Neuroscience, Surgery and Trauma, The Blizard Institute, Barts and The London School of Medicine and Dentistry, Queen Mary University of London, London E1 2AT, UK; v.lombardi@qmul.ac.uk (V.L.); y.bobeva@qmul.ac.uk (Y.B.); p.yip@qmul.ac.uk (P.K.Y.); k.schmierer@qmul.ac.uk (K.S.); 2Neuromuscular Department, Queen Square Motor Neuron Disease Centre, Institute of Neurology, University College London, London WC1N 3BG, UK; 3Translational Medicine and Therapeutics, William Harvey Research Institute, Barts and the London, Queen Mary University of London, London EC1M 6BQ, UK; j.schroth@qmul.ac.uk (J.S.); s.henson@qmul.ac.uk (S.M.H.); 4Institute of Inflammation and Aging, College of Medical and Dental Sciences, University of Birmingham, Queen Elizabeth Hospital, Birmingham B15 2TT, UK; v.pucino@bham.ac.uk (V.P.); c.mauro@bham.ac.uk (C.M.); 5Clinical Board Medicine (Neuroscience), The Royal London Hospital, Barts Health NHS Trust, London E1 1BB, UK; 6Department of Immunobiology, School of Immunology and Microbial Sciences, King’s College London, London WC2R 2LS, UK; timothy.tree@kcl.ac.uk

**Keywords:** amyotrophic lateral sclerosis, monocytes, CD11b, beta2 integrin

## Abstract

Monocytes expressing the inflammation suppressing active CD11b, a beta2 integrin, may regulate neuroinflammation and modify clinical outcomes in amyotrophic lateral sclerosis (ALS). In this single site, retrospective study, peripheral blood mononuclear cells from 38 individuals living with ALS and 20 non-neurological controls (NNC) were investigated using flow cytometry to study active CD11b integrin classical (CM), intermediate (IM) and non-classical (NCM) monocytes during ALS progression. Seventeen ALS participants were sampled at the baseline (V1) and at two additional time points (V2 and V3) for longitudinal analysis. Active CD11b+ CM frequencies increased steeply between the baseline and V3 (ANOVA repeated measurement, *p* < 0.001), and the V2/V1 ratio negatively correlated with the disease progression rate, similar to higher frequencies of active CD11b+ NCM at the baseline (*R* = −0.6567; *p* = 0.0031 and *R* = 0.3862; *p* = 0.0168, respectively). CD11b NCM, clinical covariates and neurofilament light-chain plasma concentration at the baseline predicted shorter survival in a multivariable and univariate analysis (CD11b NCM—*HR*: 1.05, CI: 1.01–1.11, *p* = 0.013. Log rank: above median: 43 months and below median: 21.22 months; *p* = 0.0022). Blood samples with the highest frequencies of active CD11b+ IM and NCM contained the lowest concentrations of soluble CD11b. Our preliminary data suggest that the levels of active CD11b+ monocytes and NCM in the blood predict different clinical outcomes in ALS.

## 1. Introduction

Amyotrophic laterals sclerosis (ALS) is, in most cases, a rapidly progressing and invariably fatal neurodegenerative disorder. The extent of upper and lower motor neuron involvement and of limb and bulbar functions impairment at the onset and in the progression of the disease conditions the significant variability in survival, ranging from less than a year (fast progressing ALS—“A-F”) to more than 5 years from symptoms onset (slow progressing ALS—“A-S”) [1]. Measures of cumulative neurological disability, such as the ALS functional rating scale revised (ALSFRS-R), and the concentration in blood or cerebro-spinal fluid (CSF) of axonal proteins, such as neurofilaments, can be used for disease stratification into homogeneous clinical phenotypes and to design smaller-size and more cost-effective clinical trials [2,3,4].

The progression of ALS has been linked to immunological dysregulation, including changes in the expression of cells of the myeloid progenitor lineage in circulation [5,6,7,8]. Peripheral monocytes represent approximately 2 to 5% of blood cells and have been recently divided into transcriptionally distinct subsets [9]. Human monocytes display remarkable heterogeneity in their surface markers expression, including CD16 and CD14, and differ in gene expression, cytokine production, homing and differentiation potential [10]. CD16-CD14+ cells are referred to as classical monocytes (CM), CD16 + CD14+ as intermediate monocytes (IM) and CD16 + CD14- as non-classical monocytes (NCM). Monocyte subsets have distinct functional properties, such as the ability to differentiate into macrophages and to promote the propagation or resolution of inflammation [9]. In neurodegenerative disorders, monocytes migrate from blood vessels into the brain, spinal cord and neuromuscular systems [11,12,13,14,15,16], where they differentiate into macrophages [17]. 

Monocytes’ accumulation in tissues where inflammation is unravelling is modulated by beta2 integrins, including CD11b, whose activated form suppresses inflammation [18]. In ALS, monocytes have been reported to have an enhanced migratory behavior and to display a transcriptional profile skewed toward a proinflammatory state, particularly in individuals with a faster disease progression (A-F) [14,15,19,20] Cytoplasmic accumulation of the nuclear protein transactive response DNA-binding protein 43 (TDP-43), a hallmark of ALS pathology, has also been described in monocytes, monocyte-derived macrophages and microglia [21]. Previous studies have shown a change in the ratio of classical to non-classical monocytes in blood from individuals with a rapidly progressing form of ALS [8]. T regulatory cells, immune cells exerting a regulatory effect on inflammation, are reduced as a percentage of parent T cells in individuals with ALS compared with NNC, a drop more pronounced in ALS cases with a more severe form of the disease [22]. Here, we study the expression in blood of monocyte subsets and anti-inflammatory active CD11b monocytes to ascertain whether changes in their peripheral expression is linked to a more favourable outcome in ALS. 

## 2. Results

### 2.1. Study Population

The demographic and clinical characteristics of the study participants are reported in Table 1. The study population included 38 ALS patients, of which 18 had fast progressing disease (A-F) and a median age at the baseline sampling (visit 1: V1) of 65.6 (range 34–86), 20 had slow progressing disease (A-S) and a median age at V1 of 62.8 (range 32–83) and 20 were non-neurological controls (NNC) with a median age at V1 of 60.4 (range 49–72). The age range and gender composition of the ALS patients and NNC were comparable. Further, 45% of the ALS patients were sampled at a second time point (visit 2: V2), of which 12 (71%) were sampled at a third time point (visit 3: V3). The median of the time intervals from the disease onset to baseline was 16.49 months (IQR 10.84–25.64) for all the ALS patients, from the baseline to V2 6.54 months (IQR 4.83–8.28) and 7.21 months (IQR 5.31–9.26) from V2 to V3. The mean ALSFRS-R score at the baseline was 35.6 ± SD 9.12 for all the ALS patients, 32.23 ± SD 10.52 at V2 and 30.67 ± SD 9.50 at V3 (Appendix A).

### 2.2. Baseline Monocyte Frequencies and Disease Progression 

Group analysis to compare the expression of monocyte subtypes at the baseline in ALS, NCM and ALS phenotypic variants did not show any significant difference. A pairwise correlation analysis (Spearman’s) showed a correlation between levels of CD11b+ NCM and ΔFRS (*R* = −0.4342, *p* = 0.0065, Figure 1A). In contrast, higher frequencies of active CD11b+ CM V2/V1 ratios were associated with reduced ΔFRS and improved survival (*R* = 0.6567, *p* = 0.0031; Figure 1B).

We also observed that high levels of HLA-DR, CD11b+ NCM and CD11b+ IM were associated with reduced survival from the baseline (*R* = −0.4588, *p* = 0.0038; *R* = −0.4203, *p* = 0.0086 and *R* = −0.3609, *p* = 0.0260, respectively), while high frequencies of baseline active CD11b+ NCM correlated with increased survival (*R* = 0.3862, *p* = 0.0166). These data suggest that the expression of NCM at the baseline is linked to a worse prognosis, while a better outcome is observed with high baseline frequencies of the same cells expressing active CD11B (Appendix A).

### 2.3. Multiple Variables Linear Regression and Survival Analysis

To study the prognostic utility of NCM, IM and CM monocyte subsets’ expression at the baseline, we fitted a linear model to estimate the effect on the disease progression rate from the onset to baseline (ΔFRS) of predefined predictor variables using multivariable linear regression analysis. To estimate the impact on survival, we utilized Cox regression in the same model (Table 2). The clinical variables included age at V1, bulbar onset of the disease, ALSFRS-R change, ALSFRS-R at V1 and gender (female). Neurofilament light-chain (Nf-L) concentrations in the plasma from the same blood samples were also considered among the independent variables. A linear regression analysis indicated that the only independent variables to significantly affect ΔFRS were ALSFRS-R at the baseline, NCM and CD11b+ NCM with estimates (unstandardized coefficients) of −0.033, 0.02 and 0.03, respectively (*p* < 0.001 for all). These data indicated that higher baseline NCM had a negative effect, while higher baseline ALSFRS-R scores a positive effect on survival (when all the other independent variables were held constant) (Table 2). In line with previous reports (22), the Cox regression analysis showed that, among the independent clinical variables, age at the baseline (*HR*: 1.06, CI: 1.02–1.10, *p* = 0.002), ΔFRS (*HR*: 1.59, CI: 1.58–9.07, *p* = 0.003) and NfL (HR: 1.059, CI: 1.0002–1.003, *p* = 0.023) were predictors of shorter survival. Higher frequencies of CD11b+ NCM were also predictors of shorter survival (*HR*: 1.05, CI: 1.01–1.11, *p* = 0.013).

The ability to predict the survival to permanent assisted ventilation (PAV), tracheostomy-free survival or last follow-up of relevant monocyte subsets was also evaluated using Kaplan–Meier (univariate) analysis. Monocyte subsets, including active and non-active CD11b cells, were dichotomized in above median (or upper tertile) and below median (or lower tertile) frequencies. In line with previous reports [23,24], higher baseline concentrations of blood Nf-L (upper tertile) were associated with a 25% reduction of survival (*p* = 0.0078; Appendix A). Above median frequencies of blood NCM and of CD11b+ NCM were associated with reduced survival (46% reduction, *p* = 0.0195 and 69% reduction, *p* = 0.0023, respectively; Figure 2A,B). In contrast, the V2/V1 ratio of activated CD11b+ CM frequencies (representing the increase of these monocytes between the baseline and V2) predicted a significant increase in survival (above median V2/V1 ratio, 65% improvement, *p* = 0.0022, Figure 2C; above median V3/V1 ratio: 48% improvement, *p* = 0.0109, data not shown).

Of all the biological markers under investigation, only the NfL blood concentrations at the baseline could discriminate patients with ALS from NNC using a receiver operating characteristic binary classification system (Appendix A, AUC: 0.9487, *p* < 0.0001). 

### 2.4. Baseline Monocyte Frequencies and Age

Higher frequencies of HLA-DR+ (*R* = 0.3850, *p* < 0.0170), of CD11b+ NCM (*R* = 0.4128, *p* < 0.0100) and of CD11b+ IM (*R* = 0.5593, *p* = 0.0003) were observed with increasing age at the baseline (Appendix A). In contrast, higher frequencies of active CD11b+ NCM (*R* = −0.496, *p* = 0.0015) were associated with younger age. Using our linear regression model to examine the effect on the expression of each of the monocyte subsets under investigation, we also showed that age at baseline independently affected the expression of active CD11b+ NCM (estimate: −0.370, CI: −0.59–0.147, *p* = 0.002), as well as of NCM (estimate: −0.24, CI: −0.43–0.064, *p* = 0.010) and of CD11b+ IM (estimate: 0.883, CI: 0.423–1.34, *p* = 0.0004). The frequencies of the same monocyte subsets did not show any correlation with age at the baseline in NNC (data not shown).

### 2.5. Longitudinal Analysis

A stability analysis showed that active CD11b+ CM was the only monocyte subset to increase in V2 compared to the baseline in ALS (red dot, zeta score > 1.5 SD) (Figure 3A) and in slow progressing individuals with ALS (A-S) (Figure 3B). In contrast, the expression of (non-active) CD11b+ CM between time points appeared stable (green dot, Figure 3A–C). 

An ANOVA for repeated measurements in a mixed model confirmed the strong up-regulation of active CD11b+ CM in V2 and V3 compared to the baseline in ALS and in A-S (Figure 3D, *p* = 0.0011 and *p* < 0.0001, respectively) in contrast with the stability across time points of CD11b+CM (data not shown).

### 2.6. Soluble CD11b and Monocyte Subsets Expression

We looked at the correlation between soluble CD11b (sCD11b) plasma concentration and the frequency of each blood monocyte subset expressing CD11b or activated CD11b in all the samples and separately in the ALS and NNC samples. Four individuals (two ALS and two NNC) had a much higher sCD11b plasma concentration compared to the other participants (>400 ng/mL) and, therefore, correlation analyses were performed with and without these outliers. Active CD11b IM and CM monocyte subsets frequencies showed an inverse correlation with sCD11b blood concentrations both in ALS (without outliers: *R* = −0.44, *p* = 0.0038 and *R* = −0.41, *p* = 0.0073, respectively, Figure 4A) and in the ALS and NNC samples combined (without outliers: *R* = −0.4690, *p* = 0.0003 and *R* = −0.4386, *p* = 0.0009, respectively, Figure 4B). The correlations between sCD11b concentrations and active CD11b+ IM and CM were also significant when the outliers were included (Appendix A). High frequencies of active CD11b+NCM showed a modest inverse correlation with sCD11b blood concentrations (*R* = −0.2833, *p* = 0.0379) only in the HC and ALS samples combined. In NNC, the sCD11b plasma concentrations did not correlate with any of the CD11b or active CD11b-expressing monocyte subset frequencies (data not shown).

## 3. Discussion

Our data indicate that the frequencies of blood NCM and CM expressing active CD11b measured at the baseline may potentially be used as predictors of survival in ALS. The proportion of active CD11b+ CM increases during the progression of ALS, a feature also appearing to be linked to a better clinical outcome (Figure 2). In contrast, in our ALS patient’s cohort, the frequency of all the monocyte subsets expressing (non-active) CD11b remains stable in the longitudinal follow-up. We also show that higher frequencies of CD11b+ NCM are associated with a reduced survival in a multivariable linear regression model and using univariate analysis (Table 2, Figure 2). In the same Cox proportional model, our study confirms previous reports that clinical parameters such as ΔFRS and NfL plasma concentrations are predictors of reduced survival in ALS [22].

Our experimental observations seem to be in line with a previous report of an acute and transient increase in a population of CD11b myeloid cells expressing HLA-DR, CD11c and CX3CR1 in ALS individuals with a slower disease progression [7]. It is also aligned to recent observations of an increase of microglia markers and macrophage activation in biofluids from ALS patients, including chitinase 1, p75 and neopterin [25,26]. A previous study from our group using unbiased proteomics of peripheral blood mononuclear cells and matched biofluids from individuals living with ALS has identified the regulation of beta-1, beta-3 and alpha-M integrins in white blood cells and plasma from ALS patients [27,28], suggesting a role for these mediators in the immunological crosstalk between peripheral circulation and inflamed brain tissue. 

CD11b, the integrin receptor for ICAM-1 and fibrinogen, facilitates the recruitment of FoxP3 + CD4 + T cells and the reduction in proinflammatory CD8+T cells in areas of inflammation [29]. Active CD11b also inhibits dendritic cells and T cell interactions [30], suppressing TLR-dependent inflammation andTh17 cell differentiation, which is known to accelerate neurodegeneration [31,32]. How the activation of CD11b receptors on myeloid cells comes into play is far from being understood. It is advocated that CD11b activation represents a genetically controlled compensatory mechanism with a regulatory effect on dendritic, monocyte and macrophage cell function, reducing inflammation and preventing the development of autoimmune responses [31].

We speculate that the conversion to an active CD11b monocyte phenotype in ALS may support monocyte/macrophage infiltration of neuromuscular and brain/spinal cord compartments, impacting positively on the inflammatory environment surrounding degenerating motor cells. In the neuromuscular compartment, this extravasation may also preserve connectivity between nerve terminals and muscles, slowing down denervation [33,34,35]. There is, nevertheless, a limit to what can be inferred, based on our investigation, from the association of circulating active CD11b monocytes and a more favorable inflammatory involvement and, ultimately, a better survival in ALS. Further investigations aimed at understanding the biology of CD11b receptors, in myeloid cells, will be required to understand the role of this integrin pathway in ALS pathobiology and to formulate potential strategies to use this molecule as a biomarker and therapeutic target for ALS.

It has been shown that the modification of peripheral monocytes and macrophages can suppress proinflammatory microglial responses in the brain, preserving neurons and prolonging life in animal models of ALS [36,37,38]. It has also been reported that blood monocytes’ gene expression is skewed towards a pro-inflammatory phenotype in ALS and, specifically, in A-F [15]. We lack a full understanding of the relation between monocytes’ molecular composition and the biological effect on the inflammatory microenvironment that accompanies neurodegenerative processes. This gap of knowledge hampers any strategy to attenuate the detrimental effects of microglia through the manipulation of myeloid cells in the periphery. A detailed transcriptional or proteomic profiling was not undertaken in our study on the blood monocyte subsets found to be linked to a better or worse clinical outcome. Our finding that age at baseline is a strong independent variable linked to higher blood frequencies of CD11b+ NCM, which, in turn, are predictors of a reduced survival in our ALS cohort, supports a possible proinflammatory shift of monocytes in ALS (Figure 2). Age is unanimously recognized as a risk factor for the development of a more severe form of ALS and is known to be associated with heightened systemic and brain-specific inflammatory responses [39].

We have also investigated the relationship between plasma concentration of the soluble form of CD11b (sCD11b) and monocyte subsets, including those with an active and non-active CD11b phenotype. Integrins such as CD11b are normally positioned on cell membranes, and their release into the extra-cellular space may result from alternative mRNA splicing, which produces polypeptides lacking a transmembrane region. Their transition into extracellular fluids may also be caused by a proteolytic cleavage from the cell surface. Soluble integrins have been reported to modulate the interaction between ligands and membrane-bound integrins affecting cell homeostasis in different pathological processes [40]. In the ALS and not the NNC samples, we observe that low sCD11b plasma concentrations are associated with high frequencies of active CD11b+ CM and IM monocytes and not with the same myeloid cells with the non-active form of CD11b integrin (Figure 4). A possible explanation of this phenomenon could be the uptake by monocyte of sCD11b from blood fluids under pathological conditions, a process possibly driving the transition of monocytes into an active CD11b state, increasing the proportion of these cells that may influence central and neuromuscular inflammation.

Our results suggest that the appearance of specific monocyte subsets in the peripheral circulation may be a prognostic indicator in a clinically heterogeneous disorder such as ALS. It also provides an indication that integrins may have a role as immunoregulatory factors that could be exploited as novel biomarkers and therapeutics for ALS. A limitation of our study is the partial understanding of the longitudinal expression of active CD11b + monocytes, given the relatively small and uneven longitudinal cohort employed in our study. Further investigations on a larger and more homogeneous longitudinal ALS population will be required to better characterize the behavior of monocytes and define their contribution to the inflammatory response that accompanies motor cell loss in the progression of ALS. It will also be important to use an unbiased approach to profile monocyte cell lineages and obtain further insight into the molecular composition of each cell type at a protein and RNA level and in different stages of the disease.

## 4. Materials and Methods

### 4.1. Participants

The blood samples used for the analysis were collected from individuals enrolled in the ALS biomarkers study (REC reference 09/H0703/27). We conducted a retrospective, single-centre study where participants and their blood samples were selected from individuals enrolled in the ALS biomarkers study between 06/2017 and 03/2020. These included ALS patients diagnosed according to the El Escorial Criteria (n:38) and non-neurological controls (NNC; n:20). ALS patients and NNC had comparable gender distribution and age at the time of baseline sampling. Exclusion criteria for ALS and NNC were systemic or organ-specific autoimmune disorders, recent treatment with steroids, immunosuppressants or immunoglobulins and recent injuries. For longitudinal analysis, additional samples were collected from a subset of ALS individuals (n:17) at two separate time points from baseline (V2 and V3). Time intervals from baseline to V1 and V3 are reported in Appendix A.

### 4.2. Peripheral Blood Mononuclear Cell (PBMC) Separation and Plasma Extraction

Plasma was separated from heparinized blood and the remaining cell component diluted with an equal amount of Dulbecco’s Phosphate-Buffered Saline. PBMC were isolated, stratifying 30 mL of diluted blood on 15 mL of lymphoprep-TM (Density gradient medium), followed by centrifugation at 2000 rpm for 40 min at room temperature (RT). PBMC’s cryopreservation for long-term storage was in 10% dimethyl sulfoxide freezing solution and foetal bovine serum. Plasma extraction was performed by centrifugation at 3500 rpm for 10 min at RT and stored at −80 °C.

### 4.3. Staining and Data Analysis

PBMC (1–2 × 10^6^ cells per tube) were stained with 100 μL of diluted viability dye zombie aqua (Biolegend^®^, San Diego, CA, USA, 1:100) for 15 min and 50 μL diluted FcBlock (Human BD) was added. Unconjugated primary antibodies (CD11b (ICRF44)-BV421, CD14 (M5E2)-BV605, HLA-DR (G46-6)-BV650, CD16 (3G8)-PE-CF594, active CD11b (CBRM1/5)-PE-Cy7) were incubated at RT for 30 min. 50 μL of brilliant stain buffer was added to reduce staining artefacts. Cells were analyzed using the NovoCyte^®^ 13-colour flow cytometer configured with 405, 488 and 640 nm lasers. The NovoExpress^®^ software was used for data acquisition and analysis. Dead cells were excluded by forward-side scatter gating and monocyte sub-classes gated using CD14/CD16 and then CD11b or active CD11b (Appendix A).

### 4.4. Measurement of Plasma Neurofilament Light Chain (Nf-L) and Soluble CD11b (sCD11b)

Analysis of Nf-L protein expression in plasma was undertaken by single molecule array (Simoa) using a digital immunoassay HD-1 Analyzer (Quanterix, Lexington, MA, USA), while soluble CD11b plasma measurement was performed by ELISA using a commercial immunoassay (ELH-ITGAM—RayBio). ALS and HC plasma samples were equally distributed on each plate and measured in duplicates.

### 4.5. Statistical Analysis

Kruskal–Wallis nonparametric test was used to compare biomarker levels across ALS patients’ subgroups and NNC. To investigate the stability of monocyte expression levels over time, we used the z-score of the means of V2/baseline ratios (log-2 transformed) to represent any increase or decrease over time (x axis) and the standard error (SE) of the same log 2 V2/V1 fold changes (Y axis) to represent inter-individual variability (monocyte sub-classes within 1.0 and 1.5 SD in the x axis and closer to 0 in the y axis showed minimal variability of expression between V2 and baseline). For longitudinal analyses, ANOVA for repeated measures fitting a mixed model with missing values at random was applied along with spaghetti plot representation. 

Pairwise correlation analysis was applied to test the association between relative percentages of monocyte subsets, functional scores and age. 

Survival was calculated as the time from baseline (V1) to permanent assisted ventilation (PAV—≥22 h/day), tracheostomy or death. Progression rate was obtained subtracting the ALSFRS-R at baseline from 48 (healthy state) divided by the time in months from reported first symptoms of disease to baseline (ΔFRS, points/month) or subtracting the last recorded ALSFRS-R score from 48 and dividing by the interval between visits in months (ALSFRS-R change, points/month). Individuals with ALS and a slower progression of the disease had a ΔFRS and ALSFRS-R change higher than - 0.5 points/month (A-S) while those with a faster disease progression had ΔFRS and ALSFRS-R smaller than −1.

A linear model was fitted with pre-defined variables already reported as predictors of the rate of disease progression and survival in ALS. Explanatory (independent) variables included gender, age at baseline (V1), bulbar onset of the disease, baseline ALSFRS-R, NfL and monocyte subset frequencies, while ΔFRS was the dependent variable. The prognostic value of these pre-defined variables and of the circulating monocytes subsets were also evaluated using Cox regression analysis, with permanent assisted ventilation (PAV) and tracheostomy-free survival from baseline as outcome. Frequencies of monocyte sub-classes were incorporated as continuous variables in the Cox prognostic and linear regression model. The ability to predict survival of relevant monocyte subsets was also evaluated using a univariate Kaplan–Meier analysis with survival from baseline. Monocyte subsets, including active and non-active CD11b cells, were dichotomized in above median or upper tertile and below median or lower tertile frequencies. Receiver operating characteristic (ROC) curve non-parametric analysis was used to discriminate between phenotypic variants and between ALS and HC based on plasma NfL concentrations. 

Continuous variables were expressed as median (interquartile range; IQR) or mean (standard deviation, SD). Monocyte sub-class frequencies were expressed as percentage of parent cells. Monocyte markers expression values were not normally distributed; hence, natural logarithm transformation was used in the analyses. Data were analysed using Prism (Version 8.0, GraphPad Software, San Diego, CA, USA) and SPSS.

## Figures and Tables

**Figure 1 ijms-23-03370-f001:**
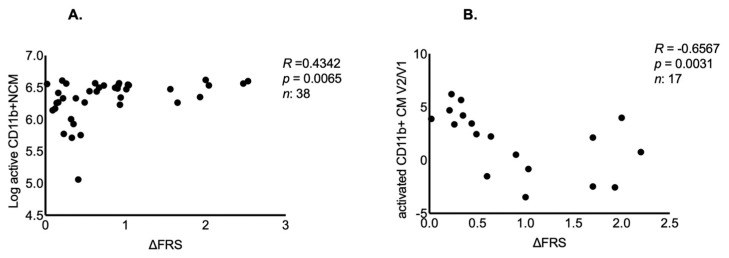
The expression of non-classical and classical monocytes and rate of ALS progression. (**A**) Higher log-transformed frequencies of CD11b+ NCM are positively correlated with ΔFRS. (**B**) The log-transformed ratio of active CD11b+ CM between V2 and baseline is negatively correlated with ΔFRS.

**Figure 2 ijms-23-03370-f002:**
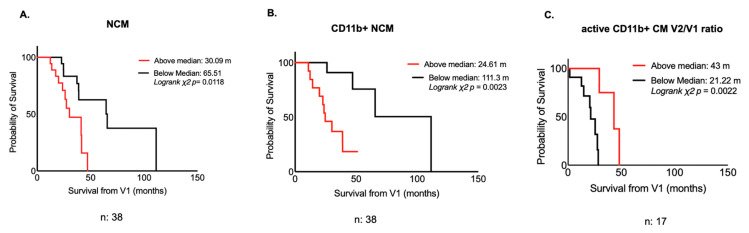
Blood monocyte frequencies and survival. Kaplan–Meier survival analysis shows a shorter survival for ALS patients with higher frequencies (above median) of NCM (*p* = 0.0118) (**A**) and with higher frequency of CD11b+ NCM, (*p* = 0.0023) (**B**). Higher V2/V1 ratios of active CD11b+ CM frequencies predict longer survival for ALS patients (*p* = 0.0022) (**C**). Red lines indicate ALS subgroups with higher analyte levels (above median), and black lines ALS subgroups with lower level (below median). Log rank chi-square and *p* values as well as survival in months calculated for each subset of ALS patients are reported for each Kaplan–Meier figure. *p*-value was obtained from log rank test chi-square. Survival is calculated from V1.

**Figure 3 ijms-23-03370-f003:**
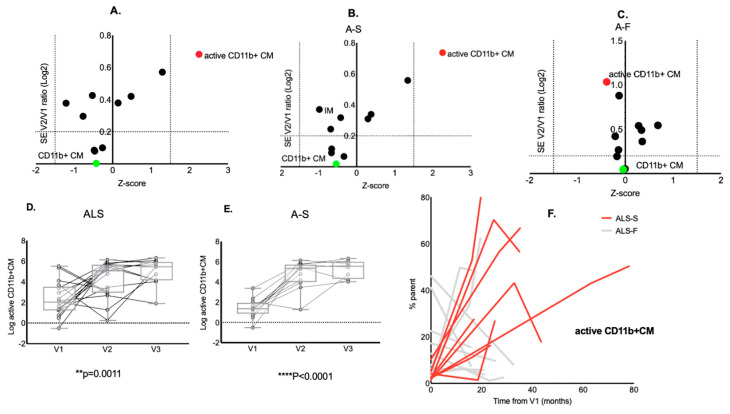
Changes in monocyte subset frequencies between longitudinal time points. Analysis of stability of expression of monocyte subsets between V2 and baseline (V1) in individuals with ALS and in slow progressing ALS (A-S). Active CD11b+ CM (red dots) are the only monocyte subset to increase over time, showing a significant elevation between baseline and V2 in ALS (**A**), in A-S (**B**) but not in A-F (**C**). In contrast, CD11b+ CM appear stable over time (green dot, **A**–**C**). Box plot representation of ANOVA analysis for repeated measures (mixed model with missing values at random) of the changes from V1 to V3 of active CD11b+ CM. In line with the results of the stability analysis, active CD11b+ CM (**F**) present an up-regulation in the later time points compared to baseline, both for ALS (*p* = 0.0011) and for A-S (*p* < 0.0001) (**D**). Spaghetti plot representing the change over time of active CD11b+ CM between V1 and V3 with a difference in the pattern of expression of A-F compared to A-S (**E**).

**Figure 4 ijms-23-03370-f004:**
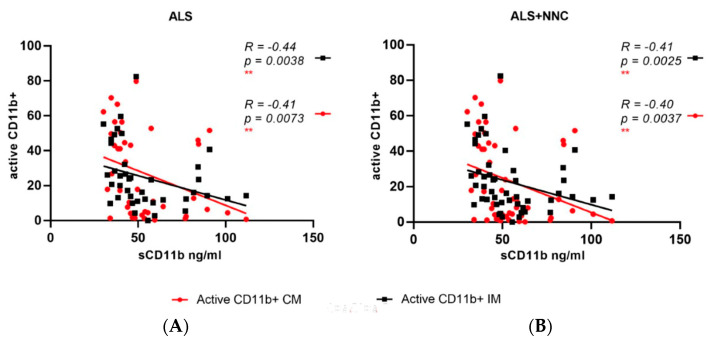
Pairwise correlation analysis between soluble CD11b plasma protein (sCD11b) concentrations and active CD11b monocytes frequencies in blood. In ALS samples only, sCD11b plasma concentrations are inversely correlated with the frequencies of active CD11b+IM (*R* = −0.44, *p* = 0.0038) and active CD11b+ CM (*R* = −0.41, *p* = 0.0043) (**A**). In all samples (ALS + NNC), sCD11b plasma concentrations show an inverse correlation with frequencies of active CD11b+IM (*R* = −0.41, *p* = 0.0025) and active CD11b+ CM (*R* = −0.40, *p* = 0.0037) (**B**). Analysis performed without 4 outliers with a sCD11b concentration >400 ng/mL (Appendix A shows a similar level of significance with all samples included).

**Table 1 ijms-23-03370-t001:** Demographic and clinical characteristics of study participants.

Clinical Characteristics	ALS (*n* = 38)	NNC (*n* = 20)
Age at baseline in years, median (IQR)	66 (11.2)	60.4 (10.8)
Female (%)	50%	50%
Site of disease onset: Bulbar (%)	44.8%	N/a
Time to baseline in months, median (IQR)	16.5 (14.8)	N/a
Baseline ALSFRS-R, mean (±SD)	35.6 (±9.1)	N/a
Baseline ΔFRS (points/month), mean (±SD)	0.8 (±0.7)	N/a
ALSFRS-R change (points/months), mean (±SD)	0.8 (±0.6)	N/a
Survival from baseline in months, median (IQR)	15.1 (12.2)	N/a
Nf-L at baseline in pg/mL, median (IQR)	106.3 (174.6)	N/a
All monocytes (%), mean (±SD)	2.7 (±2.7)	3.5 (±3.4)

*N* = total number; ALSFRS-R = amyotrophic lateral sclerosis functional rating scale-revised; ΔFRS: estimated rate of disease progression calculated subtracting the ALSFRS-R at baseline visit from 48 (ALSFRS-R approximation representing healthy neurological state) divided for time intervals in months; ALSFRS-R change: estimated rate of disease progression calculated subtracting the ALSFRS-R at last visit from ALSFRS-R at baseline, divided for the time interval in months; N/a: not applicable. Baseline: V1. NfL: neurofilament light chain. NNC: non-neurological controls.

**Table 2 ijms-23-03370-t002:** Impact on ΔFRS and survival of clinical and biological independent variable.

Covariates	* ΔFRS Estimates (95% CI)	*p* Value	** Survival HR (95% CI)	*p* Value
Gender (male)	−0.103 (−0.41–0.20)	0.79	0.83 (0.34–2.02)	0.68
Age at baseline	0.012 (−0.003–0.02)	0.15	1.06 (1.02–1.10)	0.002
ALSFRS-R at baseline	−0.033 (−0.05–0.01)	<0.001	0.93 (0.94–1.04)	0.88
Site of onset (Bulbar)	0.17 (−0.21–0.55)	0.53	0.94 (0.40–2.20)	0.90
ΔFRS	-	-	1.059 (1.58–9.07)	0.003
NfL	0.0002 (−0.0005)	0.09	1.001 (1.0002–1.003)	0.023
Active CD11b+ CM	−0.003 (0.02–0.01)	0.63	−0.013 (0.94–1.03)	0.57
Active CD11b+ NCM	0.005 (−0.018–0.028)	0.66	0.008 (0.92–1.09)	0.83
CD11b+ NCM	0.02 (0.01–0.03)	<0.001	1.05 (1.01–1.11)	0.013
NCM	0.03 (0.01–0.04)	<0.001	0.25 (0.97–1.08)	0.39

* Based on multivariate regression analysis with ΔFRS as outcome. ** Based on Cox proportional hazards model with survival from baseline as outcome. Survival time: from baseline to tracheostomy and permanent assisted ventilation. Covariates: clinical and biological independent variables used in the linear regression model. Estimates: unstandardized coefficients (how much the dependent variable varies with an independent variable when all other independent variables are held constant). ΔFRS: progression rate from onset of ALS to baseline (V1). *HR*: hazard ratios. NfL: neurofilament light chain.

## Data Availability

Anonymized data not published within this article will be made available by request from any qualified investigator.

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
