# Peer review of "The Expression of Active CD11b Monocytes in Blood and Disease Progression in Amyotrophic Lateral Sclerosis"

_ijms, 2022, doi:10.3390/ijms23063370_

Round 1

Reviewer 1 Report

In this manuscript, Yildiz et al. investigated the association between active CD11b monocytes expression and ALS prognosis. Their results showed a higher blood expression of active CD11b monocytes indicate an improved clinical outcome in ALS. The topic is important and interesting, and the authors’ findings will add new knowledge to this field. I don’t have any major concerns, only two minor suggestions:

  1. Please define A-S, A-F and HC somewhere in the introduction or table 1 legend.

  1. Please briefly introduce what is known about the different functions of CM/NCM in ALS.
  2. One of the key findings of this manuscript is the association between active CD11b monocytes and ALS prognosis. Consider to add a few sentences to discuss what is the known factors affecting the expression of active CD11b monocyte?

Reviewer 2 Report

The authors present a small sample study about the association between CD11b and survival in patients with ALS. I have the following comments to improve the manuscript:

  1. In the introduction, last paragraph, rather than stating the results, it may be more appropriate to state the aim of the study. In the paragraph before, I am missing information about the current work about studies that looked into survival and disease progression for these markers (i.e., it is unclear for the reader what this manuscripts add to what is already known or which gaps in literature this study tries to address).
  2. This sentence is unclear for the reader: "18 A-F: median age at V1: 65.6 (range 34-86) and 20 A-S: median age at V1 62.8 70 (range 32-83) and 20 age and sex matched HC...". There are many abbreviations, please introduce them appropriately. 
  3. Methods: section 4.1, where do the patients come from? Where do the controls come from? What is meant with matching, did you identify for each patient an exact match based on their gender and age, or was there a range? Please revise the word healthy control. A health control does not exist (e.g. a healthy control can be a case in another disease). The appropriate term is non-neurological control or non-ALS control. How was ALS defined, according the the El Escorial criteria? What was the timing for the second and third visit. Was this a prospective study, or did the authors retrospectively collect these data?
  4. Section 4.5 Statistical analysis. Please revise the survival analysis. By using time from disease onset you are introducing immortal time bias, meaning a patient had to survive until baseline. This can distort the actual relationship between the biomarker and survival. Please define survival as the time between the date of baseline visit (i.e. V1) and death/trach/niv or last follow-up. 
  5. Try to harmonise the language. Sometimes the authors are using V1, other times baseline. I would suggest to use baseline throughout the manuscript. 
  6. What happend to patients with deltaFRS between 0.5 and 1, they seem to fall outside slow and fast? What was the rationale for these cut-offs? 
  7. "Multivariate Cox .." line 368 should be "Multivariable Cox .. ". I am unsure why the authors categorised a perfect continuous biomarker into quartiles, or above or below median. This is a loss of information and I would suggest to just use the continuous variable in the Cox model and report the HR as appropriate. The statistical analysis section can use some restructuring, the authors are going back and forth describing different analyses and comparisons. The Survival part is in patients only, please separate that from the analysis between cases/controls. 
  8. Please remove the sentence about tests for normality; these are inappropriate (e.g. largely depended on sample size) and a visual evaluation of the distribution is beter. 
  9. Results: as in the introduction there is no rationale provided for why slow or fast progressing patients may differ, I would start the analysis and comparison between all patients with ALS vs non-neurological control. Looking at the data, this comparison will show a non-significant difference. Subsequently, I would suggest to just provide a scatter plot with deltaFRS on x-axis vs Log CD1 .. on y-axis and provide the spearman correlation. This is a more objective analysis than randomly categorising a perfectly continuously variable in order to find differences. 
  10. Figure 1B-C is inappropriate, please use the Cox model for these types of analysis to account for censoring and report the relationship between biomarker and survival as HR. 
  11. Please add the individual trajectories in Figure 2D. Is this effect not driven by the dropout of the patients with lower levels at baseline? How come that there is a datapoint in de A-S group at baseline that reaches almost -5, but that datapoint is not present in the entire ALS group?
  12. Figure 3 x-labels are not in line with what is described in statistical analysis (i.e. survival from baseline vs survival from onset). Please clarify. 
  13. Given the limitations of the study, possibly retrospective, single center (?), limited sample size and limited follow-up, I would suggest to downtune a bit the conclusion. For example, in the title: "..is associated", I think this overinterpreting the evidence and too strong for the data that the authors present. Please revise the title and abstract accordingly with the limitations outline above in mind. 

Minor:

  1. The authors use the term "ALS patients", this is seen as stigmatising by the community and the preferred term is either: "Patients with ALS", or "People living with ALS". Please revise accordingly.

Round 2

Reviewer 2 Report

The authors have addressed my suggestions sufficiently, thank you.